# Digitalizing heterologous gene expression in Gram-negative bacteria with a portable ON/OFF module

Belén Calles, Ángel Goñi-Moreno[†] & Víctor de Lorenzo[*] [ID]

## Abstract

While prokaryotic promoters controlled by signal-responding regulators typically display a range of input/output ratios when exposed to cognate inducers, virtually no naturally occurring cases are known to have an OFF state of zero transcription—as ideally needed for synthetic circuits. To overcome this problem, we have modelled and implemented a simple digitalizer module that completely suppresses the basal level of otherwise strong promoters in such a way that expression in the absence of induction is entirely impeded. The circuit involves the interplay of a translation-inhibitory sRNA with the translational coupling of the gene of interest to a repressor such as LacI. The digitalizer module was validated with the strong inducible promoters *Pm* (induced by XylS in the presence of benzoate) and *PalkB* (induced by AlkS/dicyclopropyl ketone) and shown to perform effectively in both *Escherichia coli* and the soil bacterium *Pseudomonas putida*. The distinct expression architecture allowed cloning and conditional expression of, e.g. colicin E3, one molecule of which per cell suffices to kill the host bacterium. Revertants that escaped ColE3 killing were not found in hosts devoid of insertion sequences, suggesting that mobile elements are a major source of circuit inactivation *in vivo*.

**Keywords** AlkS; Hfq; promoter; sRNA; XylS
**Subject Categories** Biotechnology & Synthetic Biology; Microbiology, Virology & Host Pathogen Interaction
**Mol Syst Biol. (2019) 15: e8777**

## Introduction

Whether naturally occurring or engineered, prokaryotic expression modules typically involve a promoter regulated by a signal-responding transcriptional factor and a downstream gene preceded by a segment encoding a 5′UTR of more or less complexity. Decoding of any gene of interest (GOI) and the making of its product is thus the result of combining translation and transcription (Blazeck & Alper, 2013; Bervoets & Charlier, 2019). That such expression modules display activated/non-activated states is the basis for a plethora of genetic circuits in which physiological or externally added signals are equalled to inputs in logic gates that deliver a connectable output ON/OFF (Wang *et al*, 2011; Moon *et al*, 2012; Brophy & Voigt, 2014; Bradley *et al*, 2016). Taken to an extreme, this can in turn be abstracted as digital 1/0 states, thereby providing the basis for ongoing attempts of biological computation and operative systems based on transcriptional factors (Ma *et al*, 2016; Nielsen *et al*, 2016; Urrios *et al*, 2016). A serious drawback of this approach is, however, that virtually no regulated expression system is really digital. While transcriptional capacities and induction rates can vary enormously among promoters, all of them—albeit to a greatly varying degree—have a measurable level of basal activity even in the absence of any inducer. This may be due to weak RNAP–DNA interactions or just be the unavoidable consequence of pervasive transcription (Creecy & Conway, 2015). One way or the other, while there are plenty of induced levels that can be properly assigned to the ON state or value 1, the OFF/0 state is often fixed by an arbitrary threshold output. This causes not only countless problems in predictability and portability of circuit performance but also makes engineering of expression systems for toxic proteins or products quite challenging (Saida *et al*, 2006; Balzer *et al*, 2013).

The literature has many cases of tightly controlled expression systems that respond to given external signals through a positive or negative transcriptional regulation mechanism. Examples include devices based on, e.g. the *tetA* promoter/operator, *araBAD* and *rha* pBAD promoters or the XylS/*Pm* expression system. They all can be modulated in a wide range of output levels from low-to-high marks (e.g. $P_{BAD}$-types and the XylS/*Pm* promoter) or medium-to-high levels of expression (e.g. the *tetA* system) while keeping relatively low basal levels of activity in the absence of inducing signal. Among these systems, the XylS/*Pm* regulator/promoter pair holds several beneficial features (Tropel & van der Meer, 2004; Brautaset *et al*, 2009). First, it is factually orthogonal, performing independently of the metabolic state of the cell. Furthermore, it is a genuine broad-host-range system and has been extensively used in various Gram-negative bacteria including *E. coli*, *P. putida*, *Pseudomonas fluorescens*, *X. campestris* and others. Induction is made by cheap benzoic acid derivatives that do not need a special transport system to enter the cell. Moreover, there is a very small likelihood of gratuitous induction or cross-talk with other expression systems and the basal level is very low, especially when combined with the

Systems Biology Program, Centro Nacional de Biotecnología-CSIC, Madrid, Spain
  *Corresponding author. Tel: +34 91 585 45 36; Fax: +34 91 585 45 06; E-mail: vdlorenzo@cnb.csic.es
  †Present address: School of Computing, Newcastle University, Newcastle upon Tyne, UK

single-copy RK2 origin of replication, which allows the industrial level production of toxic proteins (Sletta *et al*, 2004). In other cases, genetic devices have been designed for suppressing leaky basal expression levels through the engineering of super-repressors (Ruegg *et al*, 2018), exploitation of antisense RNAs (O'Connor & Timmis, 1987), or physical decoupling of regulatory elements along with conditional proteolysis (Volke *et al*, 2019). Another elegant approach has been the exploitation of diverse recombinases to maintain DNA segments in the OFF orientation in the absence of inducer (Ham *et al*, 2006), which can be fused to degradation tags to assure their transient expression, allowing the construction of synthetic gene networks capable of counting (Friedland *et al*, 2009). In this last case, expression of recombinases has to be stringently controlled in any case, and they often act in a single direction, resulting in non-reusable genetic devices. One way or the other, reliable circuits involve the use of tightly controlled expression systems. But can we really assert that the output of a given GOI in the absence of inducers is actually zero?

In this work, we have pursued the design of a general-purpose digitalizer of heterologous gene expression which combines small RNA-mediated inhibition of translation with the translational coupling of a repressor to the GOI, all framed in a double feedback circuit that could generate a switch-like regime capable of operating between OFF and ON states in a reversible manner, ruled by the presence of an externally added inducer. When placed downstream of a strong inducible promoter, the thereby resulting architecture suppresses basal expression down to altogether non-detectable levels in *E. coli* and *Pseudomonas putida*. The results below thus pave the way for generating digitalized variants of popular promoters used in synthetic circuits. Also, they allow creation of switches in which the metabolic or physiological status of cells can be entirely changed upon exposure of cells to an external signal.

# Results and Discussion

## Benchmarking low and high expression states of the tightly regulated promoter *Pm*

To evaluate the performance of the naturally occurring and tightly controlled XylS/*Pm* expression system, we adopted the standardized pSEVA238 plasmid. This is a medium-copy number vector with a pBBR1 origin of replication (30–40 copies/cell), a kanamycin resistance marker and an expression cargo composed of the *xylS* gene and the *Pm* promoter (Martínez-García *et al*, 2015; Fig 1A). The gene encoding a monomeric superfolder version of the green fluorescent protein (*msf•GFP*) was cloned downstream of *Pm* as a sensitive reporter of transcriptional activity (Fig 1A). The resulting plasmid (pS238M) was then transformed in *E. coli* CC118 strain and its behaviour analysed in individual cells by flow cytometry experiments. Figure 1B and C shows the kinetics of the expression along time at a fixed (1.0 mM) concentration of 3-methyl benzoate (3MBz) as inducer. Before induction ($t = 0$), cells show a fluorescence pattern which is very similar (but not identical) to that of the non-fluorescent control strain, i.e. *E. coli* CC118 strain transformed with pSEVA237M plasmid, containing a promoterless *msf•GFP* gene cloned exactly in the same genetic background (see grey peak in Fig 1B). Note that the median value of cells harbouring pS238M in

the absence of induction is slightly higher that the promoterless counterparts, indicating a very low but still detectable basal level (Fig 1C). The system then showed a fast response after addition of inducer as reflected in the rapid displacement of the cell population to higher fluorescence signals (Fig 1B) and the sharp increase in the median fluorescence values (Fig 1C). Most, if not all the cells, were expressing msf•GFP at 20 min after adding the inducer, and the fluorescent output increased along time to reach a plateau around 60 min later (Fig 1B and C). The level of induction increased in a dose-dependent manner, and the system was so sensitive as to detect and respond to low micromolar concentrations of inducer (Fig 1D). Population heterogeneity could also be quantified by means of the coefficient of variation expressed as a percentage (CV* 100; Raj & van Oudenaarden, 2008; Eldar & Elowitz, 2010) at each time point of the assay (Fig 1E: the higher the CV the less homogeneous the population is). The outcome of the flow cytometry experiments showed significant differences in the level of msf•GFP expressed among individual cells, especially at short times after induction, reflected in broader population peaks and higher CV values. But in general, there was a clear, apparent breach in the fluorescence readout of cells treated or not with 3MBz, as background expression of msf•GFP was very low in the absence of exogenous inducer.

## Detection and quantification of very low basal expression levels

In order to expose the level of low transcriptional activity of *Pm* which does not become apparent with GFP reporter technology, we resorted to the amplifying cascade that results from expressing a sequence-specific protease and visualizing the cleavage of a sensitive target *in vivo*. The rationale for this approach is shown in Fig 2. Plasmid pS238•NIa, which expresses the site-specific plum-pox protease NIa (García *et al*, 1989) under the same XylS/*Pm* device as before, was co-transformed in a Δ*tpiA E. coli* W3110 strain with plasmid pBCL3-57-NIa bearing an E-tagged variant of the TpiA protein, which had been engineered with an optimal cognate cleavage site. NIa is a highly active protease that efficiently cuts target proteins bearing recognition sites—even when they are grafted on a different peptidic context (Garcia *et al*, 1989). Therefore, inspection of TpiA integrity in a Western blot assay becomes an exquisite indicator of leaky protease expression. In the test shown in Fig 2, we used a protease-sensitive TpiA variant bearing the NIa target sequence at position E57 of the protein, which is known to be efficiently cleaved by the protease. The reaction can then be easily followed by means of Western blot assays using antibodies against the E-tag sequence added at the C-terminus of TpiA target enzyme. Figure 2 shows that basal expression of *Pm* in pS238•NIa plasmid was enough to produce sufficient protease as to cleave at least half of the protein produced by pBCL3-57-NIa in the absence of any inducer. This indicated that basal expression of NIa controlled by XylS/*Pm* regulatory system—which is generally invisible by other means—sufficed to trigger a biological effect. The sections below describe a feasible strategy to control this and to take non-induced gene expression to virtually zero.

## Rationale of a designer digitalizer module

Although background expression could be reduced by decreasing plasmid copy number, by using weaker RBS sequences or even by

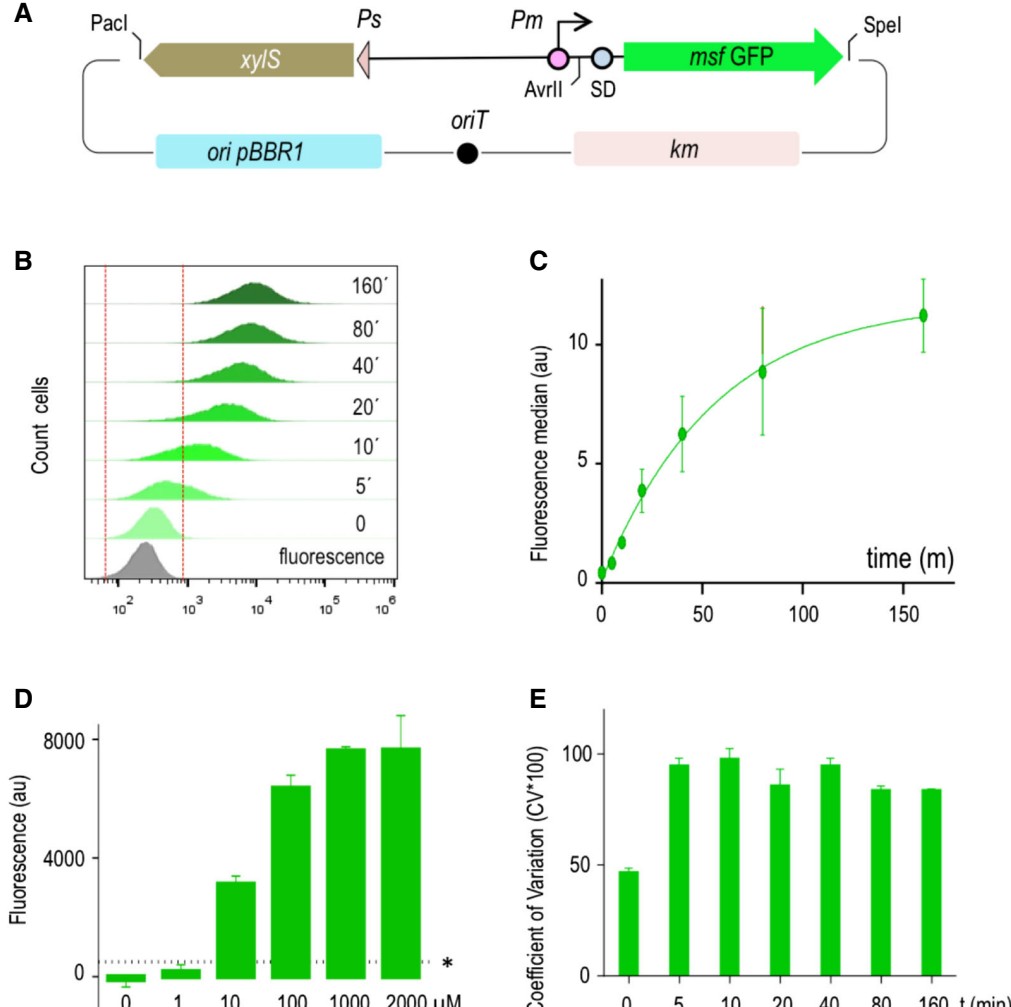

**Figure 1. Performance of a standardized XylS/*Pm*-based expression system.**

A   Schematic representation of the pS238M plasmid, harbouring an *msf•GFP* gene as reporter. Main features of the DNA backbone are indicated.

B   Evaluation by flow cytometry experiments of the basal (*t* = 0) and induced expression of GFP along time upon induction of the system with a fixed (1.0 mM) concentration of 3MBz, at the indicated time points. The same *Escherichia coli* strain was transformed with a promoterless GFP version of the otherwise identical plasmid backbone (grey plot), and this region was the considered as indicative no-fluorescence (indicated between red dashed lines).

C   Representation of the fluorescence median and SD values of three independent experiments performed as described in the previous panel.

D   Dose response of the XylS/*Pm* system. GFP fluorescence—normalized to $OD_{600}$ in all cases—was monitored with respect to the control strain transformed with the empty vector to evaluate auto-fluorescence in the absence and in the presence of increasing concentrations of inducer (3MBz) as indicated. Asterisk indicates the baseline sensitivity of the instrument. Data represent the mean and SD values of three independent experiments with eight technical replicates each performed in microtiter plates.

E   Cell-to-cell homogeneity was evaluated by means of the coefficient of variation (expressed as a percentage CV*100) of population samples analysed before and upon inducer addition at the indicated time points. Data correspond to mean CV*100 values with SD obtained from three independent experiments.

introducing mutations in the consensus promoter sequences, these approaches usually affect the induced expression level as well. Thus, we decided to explore the possibility of reducing basal levels while keeping induced levels as high as possible by introducing an additional cross-inhibition regulatory genetic circuit affecting other step of the protein expression process, i.e. mRNA translation. The designed circuit aimed at digitalizing the expression system, producing a regulatory device with a clear ON/OFF behaviour. For this reason, we named such a circuit a *digitalizer module* (see Fig 3A). The key parts of such a device include: (i)

an inducible promoter with a given level of basal expression ($P_1$), (ii) a strong, yet repressible promoter $P_2$ for transcription of a translation-inhibitory sRNA, (iii) a transcriptional repressor (R) expressed through the inducible promoter $P_1$ but translationally inhibited by the sRNA and (iv) a GOI translationally coupled (not fused) to the repressor gene. The functionality of the system stems from the fact that the repressor protein targets the strong promoter $P_2$ from which the inhibitory sRNA is produced, so that R and sRNA are mutually inhibitory. The resulting cross-inhibition between the two components is expected to result in a switch-like

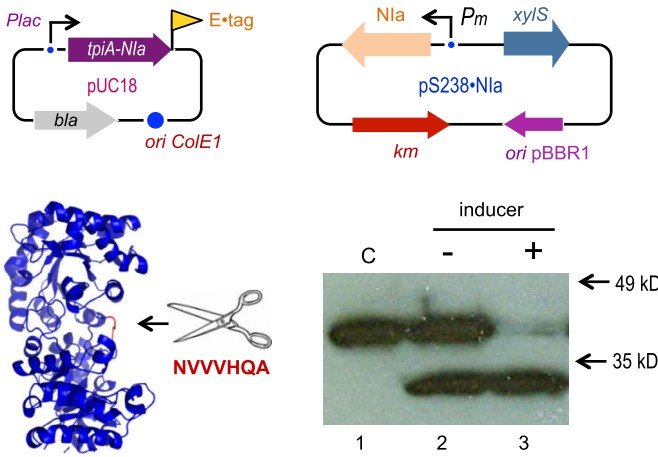

**Figure 2. Hypersensitive test of the basal activity of the XylS/*Pm* device.**

Two plasmids shown on top enabled the experimental set up used to test activity of the NIa protease when expressed through the XylS/*Pm* system. A NIa-sensitive and E-tagged TpiA variant was cloned in a high copy pUC18 plasmid, while the cognate NIa protease was expressed in a pSEVA238 vector. Analysis of the cleaving activity of the NIa protease on the target TpiA-NIa protein is detected by means of a Western blot assay with anti E-tag antibodies, in the absence (−) and in the presence (+) of 1.5 mM of the XylS/*Pm* inducer 3MBz. The control strain (C) contains pUC18 harbouring the gene coding for the TpiA-NIa protein and the empty pSEVA238 plasmid (lower panel). Note a very significant cleavage in the absence of induction.

scenario (Gardner *et al*, 2000; Kim *et al*, 2006; Lipshtat *et al*, 2006) reviewed in Zhou and Huang (2011). Qualitatively this means that if the system is in a state in which the concentration of repressor protein R is low and the concentration of the inhibitory sRNA is high then the system is OFF, as no translation can be produced. The same applies with the concentrations of R and sRNA being high and low, respectively. Thus, the system is expected to be OFF unless an external stimulus, i.e. induction of transcription, increases repressor R concentration, allowing translation to occur and causing a transition from the OFF to the ON state (Fig 3). To gain an insight into the predicted system behaviour and pinpoint key parameters for displacing the switch towards ON or OFF states, we developed the mathematical model —and run the ensuing simulations—described in Appendix Information. According to the model, the kinetic behaviour of the mutual inhibition switch as a function of the strength of each repressor (protein or sRNA) was such that the stronger the both inhibitions are, the more digital the device is; i.e., both ON and OFF expression states are more stable and the change between states is sharper (Fig 3B and C). However, a detailed analysis of the particular weight of each inhibitory element (Appendix Fig S1) showed that the strength of sRNA repression has more impact on system performance than that of the repressor R. Prediction is thus that changing binding parameters of repressor R to its cognate promoter may not have much influence on the digitalization of the system (Appendix Fig S1A) while the strength of sRNA repression is much more crucial (Appendix Fig S1B). Nevertheless, optimal digitalization is reached by combining high repression rates for both the transcriptional and the translational components of the system.

## Implementation and SBOL description of a stringent ON/OFF switch

Based on the model for a cross-inhibition switch described in the previous section, we assembled the construct depicted in Fig 4A, the main features of which go as follows. The key player of the designed post-transcriptional control circuit is a *cis*-repressing sRNA, based on a naturally occurring small transcript in *E. coli*. This regulatory sRNA is composed of two parts, a scaffold sequence and a target-binding sequence. The scaffold is provided by the MicC consensus secondary structure that has been described to recruit the RNA chaperone Hfq, which is known to facilitate the hybridization of sRNA and target mRNA as well as mRNA degradation (Yoo *et al*, 2013). The second part of the sRNA, which is replacing the natural MicC target binding region, was designed against the LacI transcriptional repressor, preventing translation of the thereby generated mRNA. This specific 24-mer fragment is the antisense sequence to the translation initiation region (starting at the very first codon) of the target repressor LacI. According to the model predictions, the sRNA ensemble was tailored to have the highest possible repression capabilities, as previously described (Balzer *et al*, 2013; Na *et al*, 2013). Synthesis of this sRNA is in turn controlled by a LacI-dependent $P_{A1/O4s}$ promoter. This synthetic $P_{lac}$ promoter derivative was selected because the kinetic parameters of RNAP–promoter interaction in combination with the position of the operator lead to the highest LacI repression factor (Lanzer & Bujard, 1988).

To ensure termination of transcription of both the bicistronic *lacI-msf•GFP* operon and the divergently expressed sRNA, two strong transcriptional terminators were added to the device. $T_{500}$, an artificial terminator derived from T82, that carries a strong hairpin (Yarnell & Roberts, 1999), was placed downstream of msf•GFP, the proxy of any GOI. Accurate termination of the sRNA, which is very important to preserve its secondary structure, was secured by means of a double synthetic $T_1/T_E$ terminator (MIT Registry, BBa_B0025). Control of the GOI in coordination with the upstream *lacI* repressor gene, which is the target of the sRNA, requires a translational coupler cassette. An efficient mechanism for coupling translation is based on the ability of translating ribosomes to unfold mRNA secondary structures. In our design, translational coupling is achieved by occluding the RBS of *msf•GFP* by formation of a secondary mRNA structure, containing a His-tag sequence added to the 3′end of the *lacI* gene. The sequence was designed so that it forms a strong hairpin ($\Delta G = -16.1$ kcal/mol) that matches the Shine–Dalgarno sequence upstream of the GOI, preventing the ribosomal recruitment and therefore translation (Mendez-Perez *et al*, 2012). In contrast, when the upstream *lacI* mRNA is actively translated, the 70s ribosome disrupts inhibitory mRNA secondary structure in the downstream gene translation initiation region thus allowing its expression. Only when the first gene is translated does the inhibiting secondary structure open up enabling translation of the second gene, as extensively documented by Mendez-Perez *et al* (2012). The whole post-transcriptional regulatory circuit was inserted downstream of the XylS/*Pm* expression module within the same pSEVA backbone (pSEVA238) as before, including the *msf•GFP* gene as reporter (Fig 4A), generating pS238D•M.

In order to ensure an accurate description of the circuit and the construct as a whole, the digitalizer was formatted using the

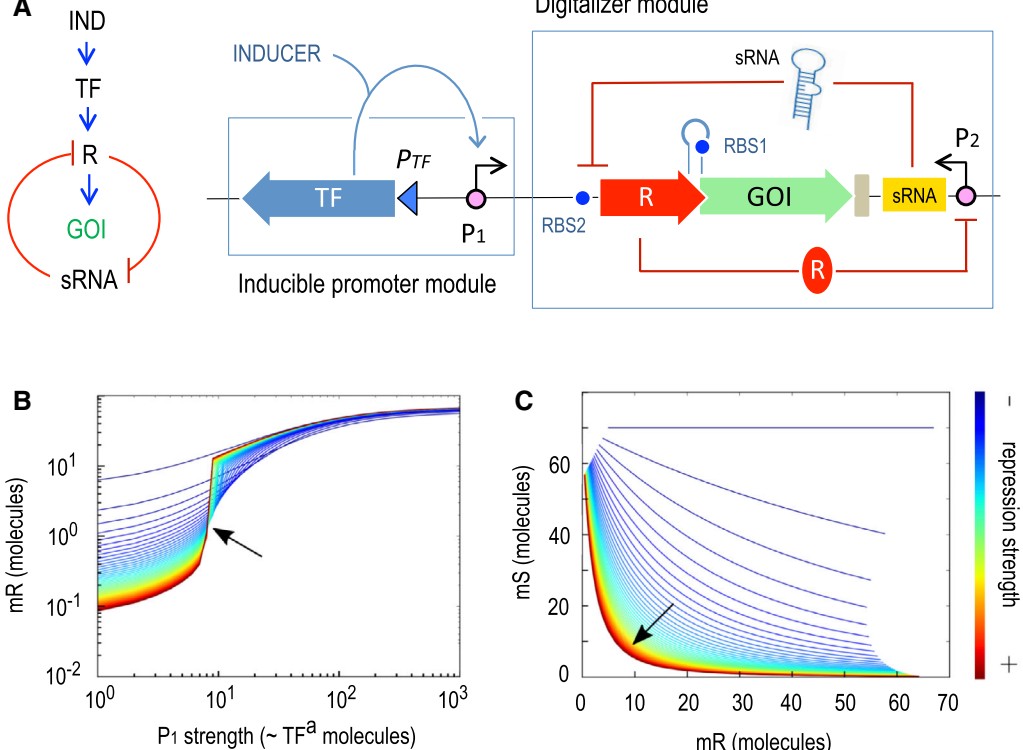

**Figure 3. Rationale and modelling of a genetically encoded digitalizer module.**

A–C (A) Expression of the GOI is controlled by a positively regulated promoter, the activity of which depends on a transcription factor (TF), and an additional switch placed downstream, consisting of a mutual inhibition circuit regulating the translation step (left). The main features of such cross-inhibition switch-like circuit, the so-called digitalizer module, include a sRNA that inhibits translation of a repressor (R) that in turns regulates the sRNA production. A translational coupler was added between the repressor and the GOI to secure a coordinated expression of both genes (right). Simulations of circuit performance are shown in panels (B) and (C). The double feedback loop formed by mR (the mRNA for both lacI and GFP) and mS (the inhibitory sRNA) is the key of the digitalization of the device. The stronger the both repressions are, the more digital the device is; i.e., both ON and OFF expression regimes are more separated (line is more horizontal) and the change between states is sharper (line is more vertical). In (B), each line is a single simulation that measures the level of mR (Y axis) while the concentration of active TF molecules—and thus P1 promoter strength—increases (X axis). The colour of the lines goes from dark red (both repressions very strong) to dark blue (both repressions very weak). The transition from OFF to ON seems to cross a single point at medium and strong repressions (arrow). This pivot point suggests the level of $TF^a$ needed to switch the system is specific and not dependent on the repression strength (except at very low values). The plot in (C) shows mS versus mR, i.e. the relationship between the two RNA species in the system. The stronger both repression effects are, the more mutually exclusive this connection is. As shown in the red line, the system is dominated by either mR or mS molecules, but not by both of them at the same time.

Synthetic Biology Open Language (Quinn *et al*, 2015; Madsen *et al*, 2019), an open standard for precise description of *in silico* biological designs and uploaded to the SynBioHub repository (McLaughlin *et al*, 2018). SBOL improves existing formats by allowing researchers to record more information on genetic circuits and share all its features in a machine-readable form (Roehner *et al*, 2016). While FASTA is used for representing plain sequences and GenBank adds the ability to annotate them, SBOL allows for specifying information such as the hierarchical structure of the circuit, its provenance and the connectivity between the non-DNA components (e.g. transcription factors) of the circuit. The SBOL design provided (Appendix Information) is focused on the hierarchical structure. The representation of the circuit was divided into two main modules: the one formed by *Pm* and downstream parts, and PA1_04S and downstream parts. Subdivision into smaller modules is shown at Appendix Fig S2. Note that a large number of tools (currently > 50) can be used to interact with SBOL designs to various extents at user's will (Appendix Information).

**Experimental validation of a digitalized expression device**

Performance of the pS238D•M construct was analysed by means of flow cytometry. Figure 4B and C shows that—as compared to the parental XylS/*Pm* system devoid of the digitalizing module—the new regulatory circuit displayed a considerable decrease in the basal levels while maintaining high expression marks at the induced state. Another important feature is that the majority of cells are effectively expressing GFP protein at earlier times ($t = 5$ min) after induction of the system (Fig 4C), indicating a faster onset of the device upon induction. Insertion of the digitalizing module also favoured cell-to-cell homogeneity as manifested in the reduction of CV values at any time along the whole course of GFP protein production (Fig 4D). Dose response of the digitalized system was also analysed under equivalent conditions as before (Appendix Fig S3). Sensitivity in this case was similar to that detected for the non-digitalized version since around 10 μM concentration of inducer was enough to trigger GFP production. Note that the total switch-like response might be affected by growth (Tan *et al*, 2009), although possible growth-

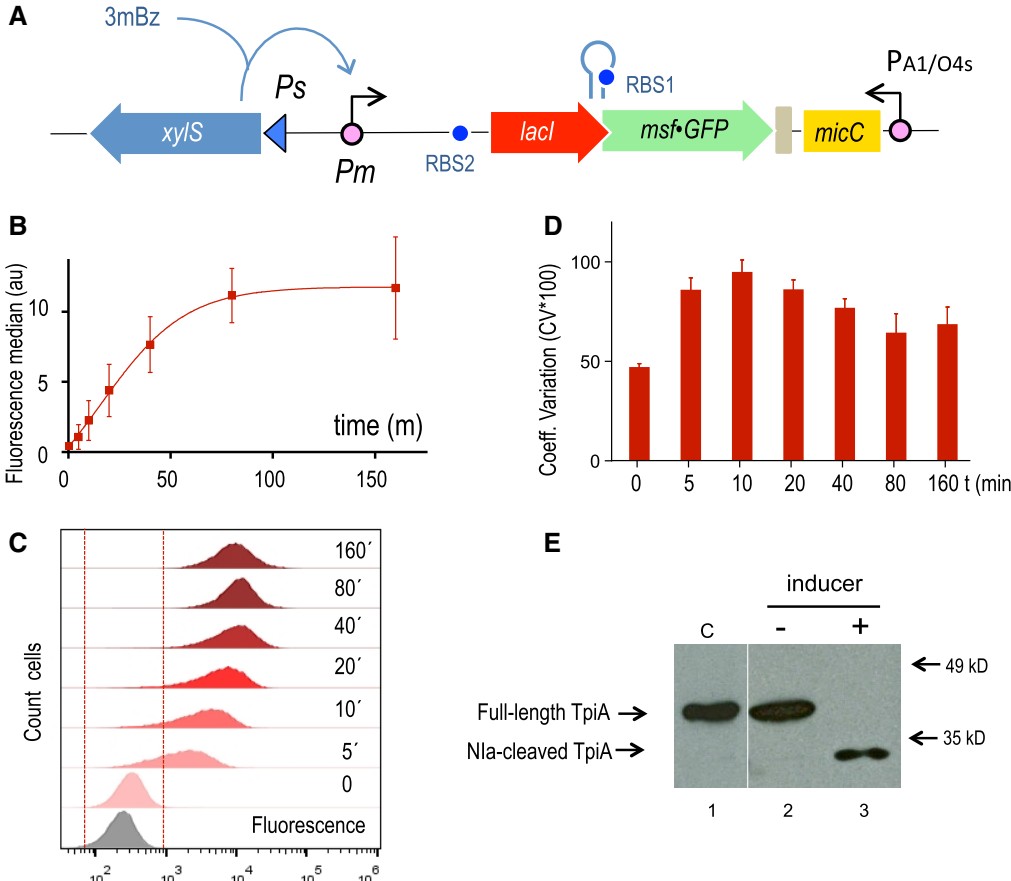

**Figure 4. Evaluation of the functionality of the digitalizer module.**

A Schematic representation of the relevant features of the specific regulatory device used to evaluate the efficiency of the digitalizer module by means of a reporter msf•GFP gene.

B GFP was used as fluorescence reporter to analyse the OFF and the ON expression profiles of the digitalized version of XylS/Pm system in flow cytometry experiments. The plot represents the median and the SD values of fluorescence from three independent experiments without inducer (t = 0) and upon 3MBz induction along time, at the indicated time points.

C Representative experiment showing the distribution of fluorescence in cell population under non-inducing conditions (t = 0) and after induction at the indicated time points (t = 5 min to t = 160 min), as indicated. The region considered as negative for the fluorescence signal is marked between red dashed lines, as assessed by control cells carrying a plasmid with a promoterless GFP (grey plot).

D Analysis of the population with respect to cell-to-cell heterogeneity by means of the coefficient of variation (percentage CV*100) at the off state (t = 0) and along the induction of the system, as described. Data correspond to the mean and SD values obtained from three separate experiments.

E The NIa protease was used as a sensitive reporter to test both the basal and the induced activity driven by the digitalized XylS/Pm device. The Western blot shows the cleavage of a NIa-sensitive TpiA target protein under the indicated conditions, which are equivalent as those used for the experiment presented in Fig 2. The control lane #1 was pasted from a different gel to show the location of the intact protein (see a detailed cleavage kinetics in Appendix Fig S6).

related effect cannot be modelled adequately due its pleiotropic effects.

In order to inspect whether the circuit borne by pS238D•M endowed expression of GOI with a degree of bistability beyond the observed switch-like, de-induction kinetics of the digitalized system was compared to that of the non-digitalized precursor device carried by pS238M. To this end, strains carrying either plasmid were induced in full with 3MBz, the inducer removed from the medium and then the decay of the fluorescent signal in the population followed over time. As shown in Appendix Fig S4, elimination of 3MBz caused a gradual loss of GFP in either strain that was indistinguishable from each other. Such a behaviour rules out any significant hysteresis in the digitalized construct and suggests the strength

of the principal promoter Pm as the only signal that rules induction and de-induction kinetics under the parameters embodied in the experimental system.

As mentioned above, simulations of the digitalizer (Appendix Fig S1) predicted that its performance depends mainly on the strength of both repressors (transcriptional and translational), being optimal when both are high. To test this prediction, we also constructed a new mutual inhibition design harbouring the classical thermosensitive variant of the very strong repressor $CI_{857}$. The sRNA was then re-configured to target the CI repressor and was placed under the control of the CI-dependent $P_{L}s1con$ promoter. This is a shortened version of the native $P_L$ promoter with additional mutations conferring a—10 consensus sequence, but retaining all three operators for

repressor binding (Gardner *et al*, 2000); see Materials and Methods section). Both $P_{lac}$ and $P_L$ promoters are controlled by negatively acting elements (LacI and CI, respectively) and repression operates in a fashion directly dependent on the rates at which the RNAP and the repressor compete for their respective binding sites. In comparative terms, the highest repression corresponds to CI-$P_L$ pair (Lanzer & Bujard, 1988). On this basis, we kept the architecture of the digitalizer module but made it dependent on the thereby described modified CI/$P_L$ module. The *cI857* gene was thus inserted under the control of the same XylS/*Pm* expression system as before, resulting in plasmid pS238D1•M. We then compared the kinetics of *msf•GFP* expression from the newly designed construct (pS238D1•M) versus the LacI-repressed version (pS238D•M). As shown in the Appendix Fig S5, the pattern of fluorescence did not change significantly along the time of the experiment suggesting that indeed the strength of CI repression did not have much influence on the system, as predicted.

Finally, to confirm the apparently complete suppression of basal expression caused by the digitalizing module we employed again the NIa protease-sensitive TpiA protein test described in Fig 2 as super-sensitive reporter of the switch tool explained above. To this end, we substituted *msf•GFP* by the NIa protease gene (giving rise to pS238D•NIa) and monitored its expression with the same two-plasmid approach as before (Fig 2). As shown in Fig 4E, while the TpiA-NIa-sensitive derivative was fully cleaved by the action of NIa under inducing conditions, proteolytic activity could not be detected in the absence of 3MBz as the inducer of the system, indicating that basal expression is negligible. This is in contrast to the result obtained without the digitalizer device (Fig 2). A detailed analysis of the kinetics of induction of NIa protease (Appendix Fig S6) revealed that transition to the ON state was fast enough to produce complete degradation of the protease-sensitive variant of TpiA in about 10 min following induction—thereby reflecting a fast expression of NIa upon switching on of the system.

## Evidence of zero expression of heterologous proteins subject to digitalized control

Based on previous results, the apparent insignificant or absent basal expression driven by the XylS/*Pm* system endowed with the digitalizing module should permit the cloning of genes coding for highly toxic products. Because some toxins are lethal to bacterial host cells at very low concentrations, any amount of leaky expression could prevent successful maintenance of these genes. Examples of such lethal proteins include colicins (reviewed in Cascales *et al*, 2007). Those included in group E have been found to destroy sensitive cells by a one-hit mechanism (Jacob *et al*, 1952), meaning that production of one single colicin molecule per cell suffices to kill the bacterium struck by the toxin. Among this type of killer proteins, colicin E3 is a specific nuclease that is active against a broad range of organisms, both *in vivo* and *in vitro* (for a review, see James *et al*, 2002). Colicin E3 is an RNase that cleaves the 16s rRNA eliminating the anti-SD sequence, completely inactivating the 30s ribosomal subunit and therefore blocking protein synthesis (Bowman *et al*, 1971; Senior & Holland, 1971; Boon, 1972). As is the case with other colicins, a few molecules of E3, even just one, are enough to kill the cell (Nomura, 1964; Maeda & Nomura, 1966; Pugsley, 1984). This makes cloning of the

*colE3* gene very unlikely even in tightly controlled expression systems, unless the corresponding ImmE3 immunity protein is co-expressed in the same cells (Mock *et al*, 1984). Not surprisingly, our attempts to clone the colicin E3 gene in the XylS/*Pm* plasmid pSEVA238 version without the cognate immune gene were not successful, suggesting that the low basal expression level of XylS/*Pm* system was categorically lethal *to E. coli*. We thus set out to test the prediction that the designed digitalized XylS/*Pm*-dependent switch could harbour genes encoding lethal proteins. To this end, the *colE3* gene had to be recloned in a streptomycin-resistant pSEVA version along with the rest of the circuit—otherwise maintaining the rest of the backbone features. This vector swapping was necessary as the *E. coli* CC118 *immE3*$^+$ strain, expressing the cognate *immE3* immunity gene, was already kanamycin resistant (Diaz *et al*, 1994). With all the right materials at hand, we used the immune *E. coli* CC118 *immE3*$^+$ strain as the host to clone the *colE3* colicin in pSEVA438 to produce pS438D•colE3 plasmid that was then transferred to *E. coli* CC118 lacking *immE3* as explained in the Materials and Methods section. Unlike before, non-immune *E. coli* CC118 cells could stably maintain the *colE3* gene cloned in plasmid pS438D•colE3. Under non-inducing conditions, cells harbouring this plasmid had a growth rate that was indistinguishable from that of the control strain, carrying the corresponding empty vector (Fig 5A). In contrast, activation of XylS/*Pm* with 3MBz led to an immediate growth arrest. The conditional killer system was very sensitive, responding to concentrations of inducer as low as 7 μM, and displaying a conspicuous digital-like behaviour (Fig 5A and B).

To further assess the utility of the system for stably maintaining the highly toxic *colE3* gene, top agar containing *E. coli* CC118 cells transformed with pS438D•colE3 was layered onto LB agar plates and exposed to the inducer 3MBz, which was impregnated in filter paper discs in the centre of the plate at increasing concentrations (Fig 5C). After incubation overnight at 37°C, visual inspection of the plates exposed production of colicin as clear halos around the spot with the inducer. The sizes of the halos of killed bacteria directly correlated with inducer concentrations and could be detected even at low micromolar concentrations, confirming the super-sensitivity of the system. These simple experiments clearly demonstrated that the plasmid containing the colicin E3 could be stably maintained unless the expression of the system is induced and that production of the highly toxic nuclease ColE3 leads to cell death.

## Roaming insertion sequences (*IS*s) are the main cause of circuit inactivation

Inspection of the growth inhibition halos of Fig 5C exposed that colonies of heterogeneous sizes survived within the conditional killer circuit implemented in pS438D•colE3 plasmid. Such *escapers* could also be indirectly detected in growth experiments in liquid medium by an increase in the optical density at late times of growth, around 12 h after induction of expression of *colE3* gene (Fig 5A). Most likely, survivors are cells that escape effects of the toxin by acquiring mutations in some of the parts involved in the implanted and/or the ColE3 targets. What could be the origin of such mutations? Insertion sequence (*IS*) elements (Siguier *et al*, 2014) are the major, sometimes dominant sources of genomic modifications in *E. coli,* accounting for near a quarter of all possible changes (Hall,

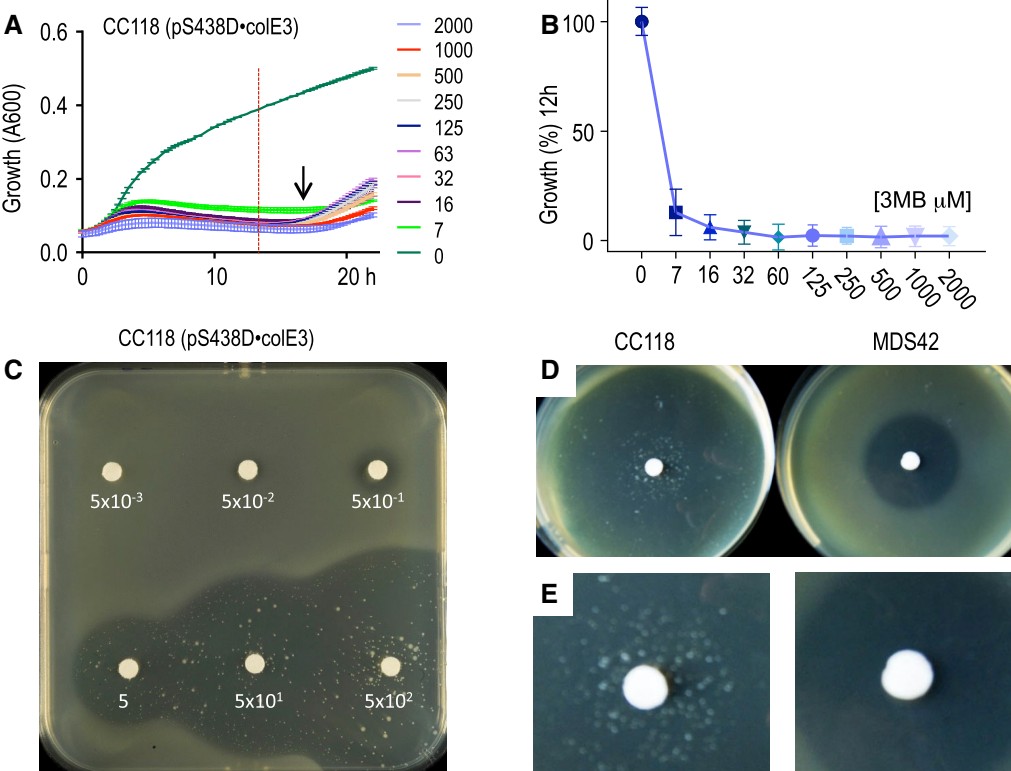

**Figure 5. The digitalizer module reduces XylS/*Pm* basal expression to virtually zero.**

A Growth curves of CC118 *Escherichia coli* cells transformed with the highly toxic *col*E3 gene cloned in a plasmid harbouring the XylS/*Pm* digitalized device described in Fig 4 without (0) or with different concentrations of inducer (3MBz 0.007–2.0 mM). The arrow indicates the approximate time when cells that escaped the action of the colicin E3. The plots represent the mean and SD growth values every 15 min, obtained from two independent experiments with eight technical replicates each. Note that the horizontal lines crossing the growth curves correspond to the limits of the vertical error bars representing the SD estimate. The red dotted vertical line is the time point selected for (B).

B Percentage of surviving cells 12 h after induction with the same range of 3MBz corresponding to the time point marked with the red dashed line depicted in (A). Growth percentage is represented as mean ± SD values from the experiments shown in (A).

C Top agar plates spread with a culture of *E. coli* CC118 cells harbouring the *colE3* gene as described in (A), induced with discs or papers soaked in 3MBz at the indicated concentrations. Halos of different size, indicating growth arrest as consequence of the toxin expression, correlate with the inducer concentration. A few survivors managed to escape the action of the colicin E3 nuclease.

D Comparison of top agar halos generated by induction of the described *colE3*-expressing construct carried by *E. coli* CC118 strain (left) or a IS-minus *E. coli* MDS42 strain (right).

E Appearance of escapers in the inhibition halos upon colicin E3 induction in the CC118 strain (left) versus the MDS42 strain (right) pinpointing *IS*s as the major source of mutations inactivating performance of the circuit.

1998). If *IS*s were the main origin of the escapers shown in Fig 5C, the use of a strain lacking them should reduce the appearance of clones unaffected by colicin E3. To test this, we transferred plasmid pS438D•colE3 to the genome-reduced strain *E. coli* MDS42 (Pósfai *et al*, 2006; Umenhoffer *et al*, 2010). This *E. coli* variant lacks 14.3% of its genome, including all *IS* elements, and thus is free of *IS*-mediated mutagenesis and associated genome rearrangements. As predicted, in contrast to previous results, clear halos devoid of surviving colonies were observed in the *E. coli* MDS42 (pS438D•colE3) strain upon colicin induction, indicating that *IS* transposition and insertion preventing colicin toxicity is the most likely source of escaper's emergence (Fig 5D and E). This inactivation mechanism is likely to affect stability of many other synthetic gene constructs (Borkowski *et al*, 2016), both prokaryotic and eukaryotic (Yokobayashi *et al*, 2002; Sleight *et al*, 2010; Wu *et al*, 2014; González *et al*, 2015). While IS elements can have both

positive and negative effects on the host (Vandecraen *et al*, 2017), they seem to have a defensive function against foreign DNA when it encodes a detrimental function (Fan *et al*, 2019).

## Suppression of high basal expression levels of the $P_{alkB}$/*alkS* device

The digitalizer module described here was designed to be compatible with any SEVA-based expression system so it could be plugged into any of them easily. Experiments with the stringently controlled XylS/*Pm* regulatory device proved the efficiency of such a module in digitalizing its performance and eliminating basal expression. The question at this point was to test whether insertion of this module into another transcriptional regulatory system with higher background expression would produce the same outcome. One of such expression devices is based on the AlkS/$P_{alkB}$ regulator/

promoter pair. This regulatory node was originally derived from the OCT plasmid of *Pseudomonas oleovorans* GPo1 (van Beilen *et al*, 1994). The positive regulator AlkS senses the presence of medium chain *n*-alkanes and induces the expression from its cognate promoter $P_{alkB}$ (Kok *et al*, 1989). In addition, the system can also be triggered by the water-soluble inducer dicyclopropyl ketone (DCPK; Grund *et al*, 1975), which is a more appropriate, non-metabolizable and less volatile inducer that do not require transport proteins to enter the cell. Similarly to XylS/*Pm*, the AlkS/$P_{alkB}$ reaches a considerable expression level when induced with DCPK (Panke *et al*, 1999; Makart *et al*, 2007). However, basal expression of AlkS/$P_{alkB}$ in the absence of inducer is about five-fold higher than XylS/*Pm*. Furthermore, the naturally occurring AlkS/$P_{alkB}$ system is subject to catabolite repression in *Pseudomonas* species (Rojo, 2009) and thus still leakier in *E. coli*. This state of affairs offered a good opportunity to inspect the ability of the digitalizer module to suppress manifestly high basal levels in a still effector-inducible system.

The plasmids assembled to compare the behaviour of the non-digitalized versus digitalized AlkS/$P_{alkB}$ are shown in Fig 6. In order to ease its analysis in *E. coli*, the reference construct (pS239M; Fig 6A) had the *msf•GFP* gene under the control of a variant of the AlkS/$P_{alkB}$ system that is independent of carbon catabolite repression. This construct is altogether identical to pS238M (Fig 1A), excepting for the parts that form the inducible expression device (AlkS/$P_{alkB}$ instead of XylS/*Pm*), thereby allowing a faithful comparison of the two systems. The basal levels of GFP expression in *E. coli* CC118 (pS239M) cells lacking DCPK were quantified with cell flow cytometry. As shown in Fig 6A, fluorescent output was relative low during the exponential and early stationary phase, but increased significantly with cell density at longer times of growth. By 24 h basically, the whole population expressed GFP at considerable levels (see dark green peak in Fig 6C). In contrast, the equivalent construct but endowed with the digitalizer module (pS239D•M; Fig 6B) virtually eliminated basal expression at any time of the experiment. This was reflected in the stillness of the corresponding peaks, which did not move towards higher fluorescence intensities even at prolonged incubation times (Fig 6B). Although the effect of the digitalizing module can be detected at any time of growth, the effective elimination of basal activity is especially evident at longer times, e.g. compare cell populations containing or not pS239D•M in Fig 6D. A sidelight of the experiments is that cells containing pS239D•M produced narrower fluorescence peaks in flow cytometry than the non-digitalized counterpart, indicating the digitalized version of the AlkS/$P_{alkB}$ system originated a more homogeneous population of expressing cells.

The behaviour of the AlkS/$P_{alkB}$ modules (digitalized or not) was then examined under inducing conditions. Expectedly, addition of saturating concentrations of DCPK (Makart *et al*, 2007) to the culture of *E. coli* CC118 (pS239M) made the population of cells expressing msf•GFP move towards higher fluorescence intensities with extended time after induction (Appendix Fig S7A). Fluorescence intensity kept accumulating in a sustained way until late stationary phase ($t$ = 24 h). As displayed in Appendix Fig S7B, cultures of *E. coli* CC118 (pS239M) had a similar expression profile. However, digitalization made cells to respond faster to the inducer (Appendix Fig S7C). A positive effect on cell-to-cell homogeneity could also be observed and attributed to the presence of the

digitalizing module, which is especially evident at early times of induction (Appendix Fig S7D).

## Hfq role in the performance of the digitalizer device

The role of Hfq RNA chaperone in the regulation of RNA metabolism has been extensively documented (Valentin-Hansen *et al*, 2004). Although some prokaryotic sRNAs can execute their functions in an Hfq-independent manner, Hfq has been shown to play crucial roles in the functionality of a large class of bacterial sRNAs (De Lay *et al*, 2013; Desnoyers *et al*, 2013; Lalaouna *et al*, 2013). Among other functions, Hfq is involved in the action of synthetic sRNAs designed for translation inhibition. Such trans-acting *hfq*-dependent sRNAs exert their function with the accessory Hfq protein that facilitates sRNA knockdown activity (Aiba, 2007). Naturally occurring MicC sRNA, whose structure has been used as part of the constructs described in the present work, has been reported to contain a scaffold sequence that recruits Hfq protein, which enhances binding and facilitates target RNA degradation (Balzer *et al*, 2013; Na *et al*, 2013). In order to test the influence that Hfq has on the action of the sRNA specifically designed for targeting the LacI mRNA of the digitalizer, we compared the expression pattern of msf•GFP from both pS239M and pS239D•M transformed in either wild-type *E. coli* BW25113 or its isogenic Δ*hfq* variant *E. coli* JW4130. Kinetics of expression of msf•GFP of both constructs in the wild-type strains was similar what had been observed in the case of having *E. coli* CC118 as the host (not shown). The absence of Hfq, however, led to poorer induction rates in all cases, probably due to the pleiotropic effect of the *hfq* mutation (Tsui *et al*, 1994). It should be noted, however, that elimination of basal expression caused by the digitalizer module was also observable in the *hfq* null strain (Appendix Fig S8). As observed before in *E. coli* CC118, basal expression of msf•GFP from plasmid pS239M in the Δ*hfq* strain was well noticeable after 24 h of growth. In contrast, GFP fluorescence intensities of cells bearing the digitalized version of pS239D•M remained as low in both $hfq^+/hfq^-$ strains as in the promoterless non-fluorescent control strains with pSEVA237M (Appendix Fig S8). This finding suggests that Hfq might not be essential when there is a high concentration of sRNA or under specific conditions whereby the sRNA-target complex can stably be formed by their own. This scenario has precedents in, e.g. the case of the RyhB, a well-established Hfq-dependent sRNA that inhibits *sodB* mRNA in *E. coli* but is also functional in the absence of the Hfq-binding region (Hao *et al*, 2011). That Hfq was dispensable for the action of the digitalizer device paved the way to use it in bacteria other than *E. coli*, as discussed below.

## Broad-host-range performance of the digitalizer module

To test the ability of the ON/OFF module to operate in different bacteria species, we first transformed *Pseudomonas putida* EM42 strain with plasmid pS239D•M and compared its behaviour with that of the parental non-digitalized plasmid pS239M. Controls included the same strain transformed with the promoterless plasmid pSEVA237M. As shown in Fig 7A and B, both in LB plates and in liquid media the basal GFP background in *P. putida* cells with the non-digitalized AlkS/$P_{alkB}$ plasmid pS239M could be easily detected by inspection of cells with blue light. In contrast, fluorescence was noticed neither in the *P. putida* strain carrying the digitalized

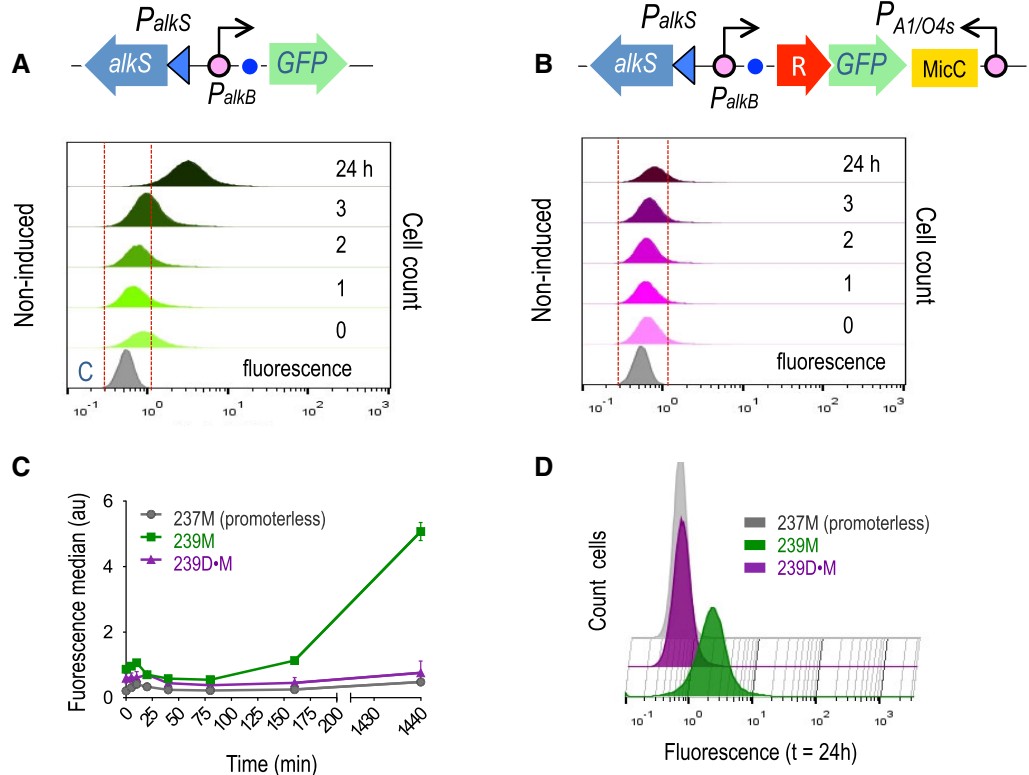

**Figure 6. Merging the digitalizer module with the AlkS/P_alkB expression system.**

A   Representation of relevant components of the positively regulated AlkS/P_alkB expression system (upper panel) and time lapse of the GFP basal expression profile analysed by flow cytometry experiments along a period of 24 h, as indicated (lower panel).

B   Assembly of the digitalized version of the same expression device (upper panel) was analysed under the same conditions.

C   Graph showing the fluorescence median values ± SD of the basal expression of the AlkS/P_alkB expression system (green line) versus its digitalized version (purple line) and the control non-fluorescent strain (grey line) along the analysed period of time. Values correspond to three independent flow cytometry experiments.

D   Comparison at late times (24 h) of the leaky expression of the AlkS/P_alkB system (green plot) and its digitalized counterpart (purple plot), which is overlapping the signal (absence of fluorescence) of the control strain (grey plot) clearly demonstrating the sharp decrease in the basal activity mediated by the digitalizer module.

Data information: In both cases (A and B), a control strain, harbouring a promoterless GFP (grey plot), was used to determine the region of no-fluorescence, comprised between the vertical red dashed lines.

plasmid pS239D•M version nor in the control strain carrying pSEVA237M. Quantitation of GFP expression of these three constructs by flow cytometry experiments confirmed this result. As shown in Fig 7C, the basal expression from the native AlkS/P_alkB device, which is considerable after 24 h of incubation, was virtually blocked by the digitalizing module as reflected by a cell population peak that remained in non-fluorescent values along time.

The performance of the digitalizing device was then tested under inducing conditions and compared to that of the non-digitalized parental system. Addition of DCPK resulted in a clear induction of msf•GFP synthesis, with a pattern resembling that observed with *E. coli* as host in both the digitalized version and its parental counterpart (Fig 7D). During the first 3 h, cell populations moved towards higher fluorescence intensities along time in a similar fashion that that observed in *E. coli*, including a faster onset in the digitalized system (Fig 7D). At longer times, however, fluorescence kept increasing in the non-digitalized construct reaching very high values after 24 h of induction (Fig 7E). In contrast, fluorescent emission of induced cells bearing pS239D•M reached a plateau after 3 h that basically remained the same for the rest of the experiment (Fig 7D and E). *Pseudomonas putida* cell-to-cell population homogeneity was also

analysed by means of the CV calculation of both non-digitalized and digitalized circuits. Results clearly indicated that the inclusion of the digitalizer device decreased population heterogeneity, as observed with *E. coli* (Appendix Fig S9). It thus seems that under some parameters, the digitalizer may make the response to the inducer faster and reduce heterogeneity, while simultaneously lower to an extent the total output level of the ON state. Although this was not captured in the model (Fig 1 and Appendix Information), it probably reflects how the dynamics of simple genetic devices can be affected by small changes in the interplay between the sRNA and its target RNA (Levine *et al*, 2007; Liu *et al*, 2011).

To further confirm the broad-host performance of the digitalizing module described in this work, we also employed the colicin E3 expressing plasmid pS438D•colE3 to transform *P. putida* KT2440 strain and run a qualitative test similar to that shown in Fig 5. The results are shown in Appendix Fig S10. A clear halo of growth inhibition appeared around the paper disc saturated with 3MBz as inducer of the XylS/*Pm* expression system, while those bacteria distant enough as to avoid inducer diffusion showed healthy growth behaviour. This confirmed that the digitalizer module prevented the toxic antibacterial colicin from expression in the absence of inducer

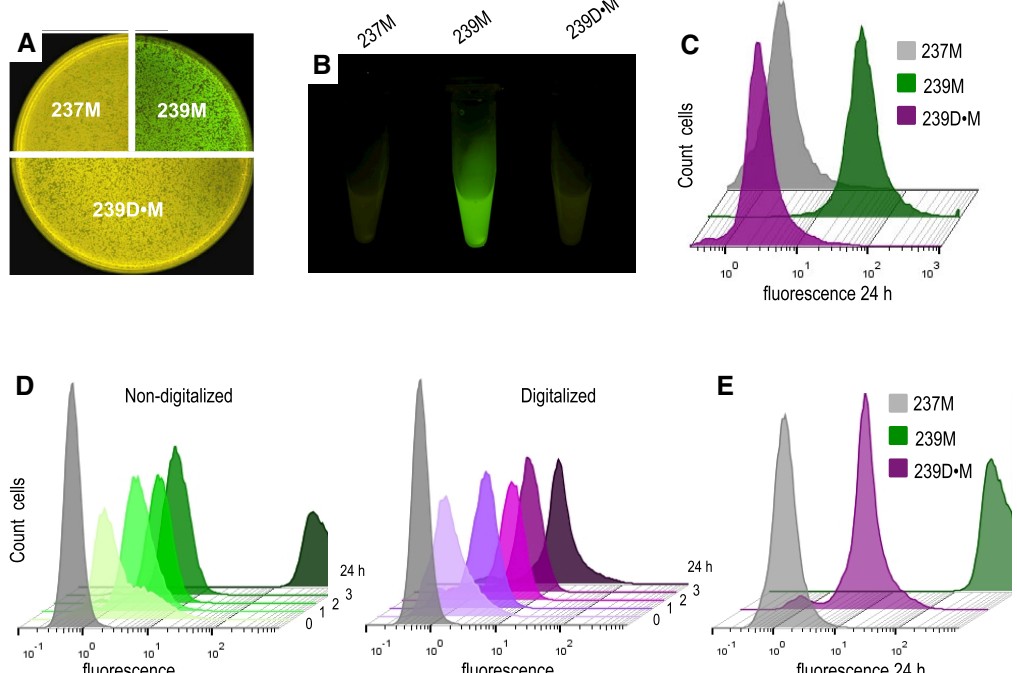

**Figure 7.  Moving a digitalized expression system to *Pseudomonas putida*.**

A, B   Inspection of basal GFP fluorescence signals in wild-type *P. putida* KT2440 cells harbouring the control non-fluorescent plasmid (237M), the default AlkS/P$_{alkB}$ expression system (239M) or its digitalized version (239D•M) in solid LB agar plates (A) or liquid cultures (B) in stationary phase of growth.

C       Non-induced cultures of cells with either construct were also analysed by flow cytometry after 24 h of incubation showing equivalent basal expression levels as those found in *Escherichia coli* cells (see Fig 6) and confirming the broad-host-range functionality of the digitalizer module.

D       Kinetics of GFP expression upon induction of *P. putida* KT2440 cells with a fixed (1.0 mM) concentration of DCPK, carrying the AlkS/P$_{alkB}$ regulatory device (left) or its digitalized version (right) at the indicated time points. The grey plot corresponds to the *P. putida* KT2440 control strain transformed with pSEVA237M (no-fluorescence).

E       Inspection of the ON state of both devices analysed above showed a reduction of the induction levels mediated by the digitalization at late times of induction (24 h).

in *P. putida* in a fashion similar to what had been shown when using *E. coli* as the host of the constructs.

### Conclusion

The use of bacterial systems as engineered machines designed to perform useful functions requires reprogramming native networks and/or constructing new genetic circuits. This task relies on the availability of optimized and well-characterized regulatory nodes that govern gene expression tightly. Control of gene expression at appropriate times is crucial, for example, to avoid metabolic burden or toxicity when using cell factories as production platforms. There is a large number of regulatory strategies but most of them are leaky and/or noisy and their use is often restricted to *Escherichia coli*. Here, we describe the rational design, mathematical modelling, computational simulation and validation of a broad-host-range digitalizing module that controls translation by means of a cross-inhibition switch-like circuit. The whole device fulfils the standards of the SEVA platform and therefore meets the plug-and-play Synthetic Biology criteria, facilitating complex gene network construction. In principle, the same design can be adapted to any positive or negative transcriptional regulation system, eliminating the major drawback of a high basal expression and allowing fine-tuning in Gram-negative bacteria, e.g. *E. coli* and *P. putida*. While there are a

number of strategies and genetic tools useful for the expression of highly toxic genes (reviewed in Saida *et al*, 2006), to the best of our knowledge successful cloning of colicin genes without the cognate immunity protein has been extremely difficult (Anthony *et al*, 2004; Bowers *et al*, 2004). In contrast, the switch tool developed in this work enables *E. coli* cells to bear the colicin E3 gene in 30–40 copy pBBR1-type vectors (Figurski *et al*, 1979; Antoine & Locht, 1992). The hereby reported results thus indicate that, by merging a tightly controlled system with a digitalizing module, expression in the absence of inducer is virtually zero. In addition, the plasmid vectors used are strain-independent and do not require special growth conditions or specific genetic backgrounds, what greatly increase the host-range applicability of the system. We anticipate that the digitalizing device will thus be instrumental to achieve phenotypes in Gram-negative bacteria that were thus far difficult to attain.

## Materials and Methods

### Bacterial strains, plasmids, cultivation conditions and reagents

The strains and plasmids used in this study are listed in Appendix Table S1. The bacterial strain used for general cloning

and construct assembly was *E. coli* CC118 except in the case of colicin E3 where *E. coli* CC118 *immE3* strain was employed (Diaz *et al*, 1994). Bacteria were routinely grown at 30°C (*P. putida*) or 37°C (*E. coli*) in LB liquid medium or in solid plates supplemented with 1.5% (w/v) agar. Shaken-flask cultivation was carried out in an air incubator at 170 rpm. When required, ampicillin (150 μg/ml), kanamycin (75 μg/ml) or streptomycin (50 μg/ml) was added. Where indicated, 3-methyl benzoate (3MBz) or dicyclopropyl ketone (DCPK) were added to the media to appropriate concentrations for induction experiments, as specified. The analysis of the expression of NIa protease was carried out in an *E. coli* W3110 Δ*tpiA* strain transformed with plasmid pBCL3-E57-NIa, expressing E-tagged TpiA protein with a NIa-cleaving site at position E57. *Escherichia coli* CC118 and *E. coli* MDS42 as well as *P. putida* KT2440 strain were employed to monitor colicin E3 expression. DNA synthesis was performed by GeneArt® (Thermo Fisher Scientific). Primers used in this work are listed in Appendix Table S2 and were provided by Sigma-Aldrich Co (St. Louis, MO, USA). Restriction enzymes and Phusion DNA polymerase used for PCR reactions were purchased from New England BioLabs inc. (Beverly, MA, USA). High-fidelity Vent DNA Polymerase was obtained from Promega (Madison, WI, USA).

### Recombinant DNA techniques

DNA manipulations were performed according to well-established methods, as described (Sambrook *et al*, 1989). Plasmid pSEVA238 bearing the XylS/*Pm* expression cargo is described in Martínez-García *et al* (2015). For the LacI-dependent digitalizing module, a DNA fragment, consisting of an *msf•GFP* gene followed by the cassette for expression of the corresponding sRNA, was custom-made by GeneArt® and provided as a DNA insert into the p57 backbone, generating plasmid pGA-LacI. The synthetic DNA was then PCR amplified from pGA-LacI by using oligonucleotides GFP-F2 and GFP-R. The DNA segment harbouring the LacI repressor was obtained from pSEVA434 plasmid, in a PCR amplification reaction using primers LacI-F and LacI-R2. All oligonucleotides contained appropriate tails to perform Gibson Assembly (Appendix Table S2). Both PCR reactions were done in a mixture containing 100 ng of the corresponding plasmid template, the necessary amount of Phusion GC Buffer 5×, 0.5 μM of each forward and reverse primers, 200 μM of deoxynucleoside triphosphate (dNTPs), 3% (V/V) DMSO and 1 μ of Phusion High-Fidelity DNA polymerase. Reaction was run with a first step of 5 min of denaturalization at 98°C, followed by 30 cycles of denaturalization (98°C, 30 s), annealing (57°C, 30 s) and polymerization (72°C, 45 s) with a final step of extension (72°C, 5 min). The amplified DNA fragments were then gel purified with NucleoSpin Extract II kit (Macherey-Nagel) and quantified. Next, a Gibson assembly procedure (Gibson *et al*, 2009) was carried out to join together LacI and the tailored synthesized fragment into the expression plasmid pSEVA238, which was previously cleaved with AvrII and SpeI restriction enzymes, producing pS238D•M. Gibson assembly conditions were as described in Aparicio *et al* (2017). The CI-dependent digitalizer module was constructed from plasmid pGA-CI, containing a synthetic DNA fragment cloned into plasmid pMK-RQ backbone provided by GeneArt®. The designed synthetic DNA segment consisted of the thermosensitive $C_{I857}$ variant of the lambda transcriptional repressor, which was edited to meet SEVA

standards, followed by the corresponding expression cassette for the cognate sRNA. Appropriate NheI and HindIII restriction sites were included between both features to later accommodate the reporter gene *msf•GFP*. The whole construct was flanked by AvrII and SpeI sites, which were used to excise it from plasmid pGA-CI and ligate it into the corresponding sites of pSEVA238 expression plasmid, generating plasmid pS238D1. Then, the *msf•GFP* reporter gene was obtained by digestion with NheI and HindIII from pS238D•M and ligated into the same sites of pS238-D1, producing pS238D1•M.

The control plasmid pS238M was constructed by PCR amplification of the *msf•GFP* gene from plasmid pGA-LacI with oligonucleotides msfGFP-HindII-F and msfGFP-SpeI-R as described for previous constructs. The amplified DNA fragment was digested with HindIII-SpeI restriction enzymes, gel purified and ligated into pSEVA238 equivalent sites. Plasmid pS238•NIa was constructed by PCR amplification of the gene encoding NIa protease from plasmid pPPVS20 (Garcia *et al*, 1989) with oligonucleotides JB653-F and JB653-R, containing SacI and KpnI restriction sites, respectively. Conditions for PCR reactions with Vent DNA polymerase were similar to those before. The resulting PCR product was cleaved with flanking enzymes SacI and KpnI, gel purified and ligated with pSEVA238 plasmid digested with the corresponding enzymes. Cloning of NIa protease gene into the digitalized pSEVA238 derivative was done following the same procedure and PCR conditions. First, the *nIa* gene was PCR amplified with oligonucleotides NIa-F-SR syst, containing an appropriate NheI restriction site to couple its expression with the upstream transcriptional repressor LacI, together with primer JB653-R, bearing a KpnI site at the 3′end. The resulting DNA segment was inserted into NheI-KpnI sites of pS238D•M, replacing the *msf•GFP* gene and producing pS238D•NIa. For assembling plasmid pS438D•colE3, the whole digitalizing module was first excised from pS238D•M by digestion with AvrII and SpeI and ligated to the same sites of pSEVA438, originating pS438D•M. In parallel, the *colE3* gene was PCR amplified with high-fidelity Vent DNA Polymerase from plasmid pEDF5 (Diaz *et al*, 1994) with oligonucleotides colE3-F and colE3-R (which deliver flanking NheI and EcoRI sites). The resulting *colE3*-containing fragment was then inserted into NheI-EcoRI sites of plasmid pS438D•M in substitution of the *msf•GFP* gene, leading to plasmid pS438D•colE3. pSEVA239M was constructed by ligating the HindIII-SpeI fragment excised from pSEVA429 into the same sites of plasmid pSEVA237M. Restriction of pSEVA239M with AvrII-Spe rendered two DNA fragments, i.e. the *msf•GFP* gene and the pSEVA239 backbone. The latter was isolated from an agarose gel, purified and ligated the to the digitalizing LacI-dependent module obtained from plasmid pS238D•M with the same procedure, leading to plasmid pS239D•M. In all cases, the structure of every construct containing DNA segments obtained by PCR amplification was first verified by colony PCR and later confirmed by restriction mapping and DNA sequencing (Secugen Sequencing and Molecular Diagnostics, Madrid, Spain).

### NIa protease expression analysis by Western blot experiments

Cells transformed with specific plasmids were grown in LB medium at 37°C overnight as described above, diluted to an $OD_{600}$ of 0.05 and incubated again until a $OD_{600}$ of 0.4, at which point expression of the NIa protease was induced by addition of

3MBz (1.5 mM) for 3 h. Then, sample cells of equivalent $OD_{600}$ were harvested by centrifugation at 3,773 $g$ for 10 min at 4°C. Cell pellets were then resuspended in 50 µl of SDS-sample buffer (120 mM Tris-HCl pH 6.8, 2% w/v SDS, 10% v/v glycerol, 0.01% w/v bromophenol blue and 2% v/v 2-mercapto-ethanol). Samples with thereby prepared extracts equivalent to ~$10^8$ cells per lane were loaded on SDS–PAGE gels (Miniprotean system, Bio-Rad). Following electrophoresis, gels were transferred to polyvinylidene difluoride membranes (Immobilon-P, EMD Millipore, Billerica, MA, USA) using semi-dry electrophoresis transfer apparatus (Bio-Rad Laboratories). Membranes were next blocked for 2 h at room temperature with MBT buffer (0.1% Tween and 5% skimmed milk in phosphate-buffered saline, PBS), washed three times with PBS-T buffer (PBS 1× supplemented with 0.1% Tween) and incubated with 1:1,500 dilution of anti-Etag antibodies (Phadia) in PBS-T buffer for 1 h at room temperature. Blots were washed three times with the same buffer to remove the excess antibody. The membrane was next incubated another hour with an anti-mouse IgG antibody conjugated with peroxidase (Sigma) diluted 1:5,000 in PBS-T, rinsed three times with the same buffer and then soaked in BM chemiluminescence blotting substrate (Roche Molecular Diagnostics). After 1 min of incubation in the dark, the blots were detected with X-ray films. Experiments were replicated at least three times.

### Top agar experiments

Cultures of the bacteria indicated transformed with pS438D•colE3 were grown overnight with vigorous shaking in LB medium. Cell samples were then collected and centrifuged at 3,773 $g$ in a tabletop centrifuge during 5 min, and the pellets resuspended in PBS to an $OD_{600}$ of 0.5. Then, 100 µl of sample cells was mixed with 4.5 ml of top agar [0.7% agar supplemented with 0.9% (w/v) NaCl] that had been melted at 50°C. The mixture was poured into LB agar plates. 3MM Whatman paper discs soaked in 3MBz at the indicated concentrations were placed on the solidified layer, and plates were incubated overnight at the adequate temperature. Experiments were performed three times with two technical replicates each.

### Growth curves

*Escherichia coli* CC118 cells transformed with plasmid pS438D•colE3 were grown overnight in LB medium and then diluted 1:20 in 96-multiwell microtitre plates (NunclonTM Delta Surface; Nunc A/S, Roskilde, Denmark) containing 190 µl LB medium, supplemented with streptomycin. Cells were then grown with rotatory shaking in a SpectraMax M2e plate reader (Molecular Devices, LLC., Sunnyvale, CA, USA) for 3 h. Induction of *colE3* gene was triggered by addition of the indicated 3MBz concentrations (0 to 2 mM) and the incubation at 30°C continued up to 24 h. Bacterial growth was recorded as the optical density at 600 nm measured every 15 min. The data shown correspond to two independent experiments with eight technical replicates each.

### Flow cytometry analysis of cells bearing the digitalizing module

For quantification of GFP expression in flow cytometry experiments, specific strains containing the relevant constructs were inoculated into filtered LB medium and grown to stationary phase. Cells that had been grown overnight were then diluted to an $OD_{600}$ of 0.05 in fresh-filtered LB and incubated at the appropriate temperatures to an $OD_{600}$ of 0.4, when the corresponding inducer was added, as described. This time point was considered as $t = 0$. Following exposure to the inducer, 1 ml samples were harvested at various time points, as indicated in the text, and spun down in a tabletop centrifuge at 11,337 $g$ for 1 min. Then, cells were washed in 500 µl of filtered PBS, centrifuged again, resuspended in 300 µl of 0.4% (W/V) paraformaldehyde and incubated at room temperature for 10 min. After cell fixation, cells were washed twice in 500 µl of filtered PBS and finally resuspended in 600 µl of filtered PBS and stored on ice until analysis. The final $OD_{600}$ was adjusted to a value < 0.4 in all samples. Single-cell fluorescence was then analysed by flow cytometry using a GALLIOS cytometer (Perkin Elmer) or with a MACS-QuantTM VYB cytometer (Miltenyi Biotec GmbH). GFP was excited at 488 nm, and the fluorescence signal was recovered with a 525/40 nm band pass filter. At least 25,000 events were analysed for every aliquot. The GFP signal was quantified under the experimental conditions tested by firstly gating the cells in a side scatter against forward scatter plot, and then, the GFP-associated fluorescence was recorded in the FL1 channel (515–545 nm). Data processing was performed using the FlowJo™ software as described elsewhere (www.flowjo.com). All experiments were carried out at least three times with two technical replicates.

### Monitoring fluorescence in populations grown in microtitre plates

Dose–response experiments (Appendix Fig S3) were run by inoculating overnight cultures of the corresponding non-digitalized, digitalized and control strains, diluted 1:20, into a total volume of 200 µl of LB+Km in 96-well microtitre Costar black plates with clear bottom (Thermo Fisher Scientific Inc., Pittsburgh, PA, USA). The cultures were incubated at 30°C with rotatory shaking in a Victor-2 multireader spectrophotometer until mid-exponential phase, when expression of GFP was induced with increasing concentrations of 3MBz and incubated again until late stationary phase. Growth and GFP fluorescence were simultaneously recorded every 15 min along the whole experiment. System deactivation after inducer depletion (Appendix Fig S4) was also inspected with a Victor2 plate reader. To this end, overnight cultures containing the control, the non-digitalized and the digitalized strain, were inoculated in LB supplemented with Km at an $OD_{600}$ of 0.05 and grown until it reached ~0.4, at which point expression of GFP was triggered by addition of 3MBz (1 mM) followed by incubation for three additional hours. During this period, samples were collected at the times indicated for monitoring growth and GFP expression. Cells from the fully induced cultures were then spun down 10 min at 3000 g in a centrifuge 5810 R (Eppendorf) and washed twice with fresh LB supplemented with Km for eliminating inducer. Cell suspensions were then diluted five-fold to adjust their $OD_{600}$ to ~0.4, 200 µl samples deposited in Costar black plates with clear bottom and incubated for 4 h, during which growth and GFP were simultaneously monitored every 15 min. At that point, cultures were re-diluted five-fold with fresh LB medium plus Km and incubated again during 4 additional hours with $OD_{600}$ and green fluorescence recorded as before.

**Mathematical modelling and SBOL descriptions**

Mathematical modelling of the digitalizer device was done for analysing the dynamical features of the system at different rate values and simulated parameters. Simulations were coded in Python. The performance of the switch was captured by the following set of ordinary differential equations (Miró-Bueno & Rodríguez-Patón, 2011) as explained in detail in the Appendix Information. For generating a SBOL version of the circuit of interest, we used iBioSim (http://www.async.ece.utah.edu/ibiosim) and BioCad (http://biocad.io). A version of the SBOL design with hierarchical information is available at SynBioHub (McLaughlin *et al*, 2018) under the URL: https://synbiohub.org/public/Digitalizer/Digitalizer_collection/1.

# Data availability

The design built in this work is available from the following resource: SBOL representation of the digitalizer: SynBioHub, https://synbiohub.org/public/Digitalizer/Digitalizer_collection/1

**Expanded View** for this article is available online.

## Acknowledgements

Authors are indebted to Eduardo Diaz (CIB, Madrid) for generously sharing valuable materials. This work was funded by the SETH Project of the Spanish Ministry of Science RTI 2018-095584-B-C42, MADONNA (H2020-FET-OPEN-RIA-2017-1-766975), BioRoboost (H2020-NMBP-BIO-CSA-2018) and SYNBIO4-FLAV (H2020-NMBP/0500) Contracts of the European Union, the S2017/BMD-3691 InGEMICS-CM funded by the Comunidad de Madrid (European Structural and Investment Funds) and the SynBio3D project of the UK Engineering and Physical Sciences Research Council (EP/R019002/1).

## Author contributions

BC designed and run the experiments shown in this work. AG-M performed computational models and run *in vivo* simulations of the digitalizer module. VdL provided direct supervision, interpreted results and helped design this research. BC, AG-M and VdL wrote the manuscript. All authors read and approved the final manuscript.

## Conflict of interest

The authors declare that they have no conflict of interest.

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
