## [Review Process File · Molecular Systems Biology]

Digitalizing heterologous gene expression in Gram-negative bacteria with a portable on/off module

Belén Calles, Angel Goñi-Moreno, and Víctor de Lorenzo

Review timeline:

Submission date:	17 March 2019
Editorial Decision:	11 May 2019
Revision received:	23 August 2019
Editorial Decision:	27 September 2019
Revision received:	16 October 2019
Accepted:	24 October 2019

Editor: Thomas Lemburger

Transaction Report:

1st Editorial Decision

11 May 2019

Thank you again for submitting your work to Molecular Systems Biology. We have now heard back from the three referees who agreed to evaluate your manuscript. As you will see from the reports below, the referees find the topic of your study of potential interest. They raise, however, several points that we would ask you to address in a revision of this work.

The major points addressed by the reviewers refer to the following issues:

- experiments should be conducted to show whether the circuit displays hysteresis
- dose-response curves should be shown for the digitalizer circuit.

Reviewer #3 makes also the suggestion that the circuit on/off ratio could be further optimized for *P. putida*. While we would certainly welcome data on this point, we feel that inclusion of these experiments is not mandatory for publication of the present manuscript.

If you feel you can satisfactorily deal with these points and those listed by the referees, you may wish to submit a revised version of your manuscript. Please attach a covering letter giving details of the way in which you have handled each of the points raised by the referees.

Thank you again for the opportunity to consider your work and we look forward receiving your revised manuscript.

 REFEREE REPORTS

Reviewer #1:

This manuscript evaluates the possibility of building a bacterial Digitizer gene circuit that lacks basal expression. The system is based on XylS/Pm, a system inducible by 3MBz, for example. The original system's basal expression is leaky, as verified using TpiA. Even one cellular copy of the Nla protease cleaves TpiA, which is detectable by western blotting. Next, to build a truly leak-free system, a toggle switch-like motif is tested using mathematical modeling predictions. Specifically, a repressor co-transcribed with the target gene represses a small RNA's expression, while the small RNA inhibits the repressor. Mathematical modeling suggests that the leaky expression can vanish completely. The efficiency of the system is better if the repressor and small RNA are more efficient. An experimental version of the system using the LacI repressor and the micC sRNA indeed eliminates basal expression, as tested with Nla cleaving TpiA, and then by expressing Colicin E3 that kills *E. coli*. This revealed a gene circuit inactivation, mainly by IS-sequence insertions, as confirmed using an IS-free, engineered strain. The robustness of the new circuit design is tested by replacing LacI with AlkS and doing the corresponding promoter switch. Moreover, the digitizer design seemed to work in an Hfq-free strain and another bacterium as well, further supporting its robustness.

These are interesting systems-level designs and important discoveries, with multiple aspects tested. The work deserves to be published, once the following comments can be addressed.

- (1) Toggle-switch like bistable systems have hysteresis - meaning that the stimulus needed to switch them one way is different from the stimulus needed to switch them back. Is this true for the digitizer? How would it affect its response speed to alternating On/Off conditions?
- (2) Besides Hfq-dependent sRNA-s, there are Hfq-independent ones. This could be discussed.
- (3) The secondary mRNA structure between the repressor and the target gene is an important part of the design. This should be more emphasized. How could it be modeled? How would the stability of the hairpin affect circuit performance?
- (4) It would be important to show a full dose response for the digitizer circuit in Figure 4, equivalent to Figure 1 panel D.
- (5) Considering the rightward shift of the histogram in Fig. 4C at 5 minutes, and that the CV is more sensitive to a right tail than to the overall width of a distribution, it would be important to plot histograms and calculate CVs before 5 minutes post-induction if technically possible. The CV may reach a peak before 5 minutes.
- (6) Figure 7 - could dose-responses and CVs be shown, equivalent to Figure 1?
- (7) Earlier gene circuit designs - such as ones based on recombinases, see PMID:19478183 - should have zero leak. The pros/cons of these systems should be mentioned and discussed.
- (8) Could something similar be implemented in eukaryotes?
- (9) Apparently, the gene circuit can degrade by IS-insertions. Other work on gene circuit evolution would be useful to mention, such as PMID:26324468.

Reviewer #2:

Calles et al constructed a feedback system that permits the expression at high as well as at virtually zero expression levels, a system they term "digitalizing". The authors explored the system behavior with different reporters and cells to show the robust behavior of this system. In particular, they tested various reporters: GFP, protease based amplification and the highly toxic *colE3* gene. Furthermore, they have shown the applicability with two induction systems, 3MB and DCPK. Some surviving colonies can arise even under *ColE* inducing conditions; they show that this is due to the mutations caused by transposons and a strain lacking transposons can be used to eliminate these colonies. The system can be also expressed in *Pseudomonas*.

Minor comments:

1. The authors use system with a net positive feedback loop, which can result in a switch-like or bistable response, depending on whether corresponding the open-loop response is ultrasensitive or not. The term "digitalizing" is reasonable to combine / summarize these two options (a switch-like or bistable). However, if the authors want also to use the term bistable for the system then a hysteresis experiment should be performed with IPTG [PMID: 14973486], taking into account that the basal expression has a major effect on the bistable parameter domains [PMID: 26599573].
2. The authors use CV to assess variability, which is typically defined as the SD/mean. Therefore, they should specify that they use CV*100, CV percentage [%].
3. Some of the sentences should be spell-checked or clarified for better understanding. Examples:
 "Whether naturally-occurring or engineered prokaryotic" comma before prokaryotic
 "The highest the CV the less homogeneous the population is." higher
 "Since just one N1a molecule per cell can cleave the target," meaning should be clarified.

Reviewer #3:

In this manuscript, the authors reported a novel circuit design to achieve zero-expression of protein which plays a digital-like behavior. They rationally designed a simple digitalizing module controlling translation by means of a bistable circuit to overcome the leaky basal expression of inducible promoters. After the ON/OFF ability of the switch module tested, it was validated with different promoter devices and showed compelling performance in different bacteria species. The work is a good example of what rational design combining proper selection of biological part can achieve. The digitalizer module shows potential for practical applications such as toxic protein production. The module represented in SBOL and fulfilling the standards of SEVA-based platform make it a good toolkit for future synthetic biology researches and applications.

There are a few points I would suggest:

- 1) The authors mentioned that cells with digitalizer module showed obviously faster response after the induction of the system than non-digitalizer. The authors should make a further discussion about the mechanism behind this phenomenon.
- 2) The authors used fixed inducer concentration in the validation experiments of the digitalizer module with GFP as reporter. A dose-response study of the digitalizer module would be more systematic.
- 3) In Figure 7D, it is interesting to notice that the on/off ratio of digitalizer module (pS239D•M) in *S. putida* is not as good as in *E. coli*. In *S. putida*, and there is a gap in on/off ratio between digitalizer and non-digitalizer. Is it because that the repression from LacI repressor is weaker in *S. putida*? Can the performance be improved by changing the repressor? It would be helpful if the authors could address this issue.
- 4) The presentation of some figures can be improved. Figure 1E and 4D are missing time(t) for the last column. Different fonts were used in one figure (Figure 3). Inconsistent graphic styles of subfigures (Figure 1D and 1E, Figure 4B and 4D) do not harmonize.

1st Revision - authors' response

23 August 2019

Reviewer #1

(1) Toggle-switch like bistable systems have hysteresis - meaning that the stimulus needed to switch them one way is different from the stimulus needed to switch them back. Is this true for the digitalizer? How would it affect its response speed to alternating On/Off conditions?

The *digitalizer* module is not a real *toggle switch* and therefore bistability cannot be taken for granted. We have addressed this question in the revised ms. by carrying out experiments (now in Appendix Fig. S4) for examining the de-induction kinetics of the XylS/Pm device with and without the digitalizer module. The results clearly indicate that there is no hysteresis and the ON and OFF states of the system are exclusively dependent on the inducer. This makes the construct to be a most useful digital, bimodal device that quickly resets to basal levels when the inducer is removed. We have scanned the text of the article for eliminate any possible ambiguity in this respect.

(2) Besides Hfq-dependent sRNA-s, there are Hfq-independent ones. This could be discussed.

A brief discussion about the Hfq-independence of the sRNA has been added to the text.

(3) The secondary mRNA structure between the repressor and the target gene is an important part of the design. This should be more emphasized. How could it be modeled? How would the stability of the hairpin affect circuit performance?

The structure and stability of the optimized translational coupling used in this work was described in detail by Mendez-Perez and colleagues in the paper cited in this work. For clarification, we have introduced some additional info in the revised article.

(4) It would be important to show a full dose response for the digitizer circuit in Figure 4, equivalent to Figure 1 panel D.

The dose-response graph for the digitalized circuit (and comparison with the non-digitalized counterpart) has been included in Appendix Figure S3.

(5) Considering the rightward shift of the histogram in Fig. 4C at 5 minutes, and that the CV is more sensitive to a right tail than to the overall width of a distribution, it would be important to plot histograms and calculate CVs before 5 minutes post-induction if technically possible. The CV may reach a peak before 5 minutes.

Interesting point! Alas, our experimental platform does not allow take and process samples in less than 5 minutes after induction. Yet, we argue that this is still significant!

(6) Figure 7 - could dose-responses and CVs be shown, equivalent to Figure 1?

CVs values for these constructs are now shown in a new Appendix Figure S9

(7) Earlier gene circuit designs - such as ones based on recombinases, see PMID:19478183 - should have zero leak. The pros/cons of these systems should be mentioned and discussed.

Circuit designs based on recombinases have been included and briefly discussed in the Introduction of the revised ms.

(8) Could something similar be implemented in eukaryotes?

In principle, a similar digitalizer module could be implemented in eukaryotes, provided that orthogonal and appropriate parts, following the principles underlying the circuit, are used. Relevant components could eventually derive from prokaryotic organism. Actually, many prokaryotic devices (i.e. the tetracycline responsive regulatory system) have been applied to control gene activity in eukaryotes. But at this point this is a mere speculation.

(9) Apparently, the gene circuit can degrade by IS-insertions. Other work on gene circuit evolution would be useful to mention, such as PMID:26324468.

A few extra sentences re IS vs. circuit stability have been entered in the revised ms.

Reviewer #2

1. The authors use system with a net positive feedback loop, which can result in a switch-like or bistable response, depending on whether corresponding the open-loop response is ultrasensitive or not. The term "digitalizing" is reasonable to combine / summarize these two options (a switch-like or bistable). However, if the authors want also to use the term bistable for the system then a hysteresis experiment should be performed with IPTG [PMID: 14973486], taking into account that the basal expression has a major effect on the bistable parameter domains [PMID: 26599573].

Correct! See response to remark (1) of Reviewer #1 above

2. The authors use CV to assess variability, which is typically defined as the SD/mean. Therefore,

they should specify that they use $CV \times 100$, CV percentage [%].

We have clarified throughout the text that we present CV values as percentages ($CV \times 100$)

3. Some of the sentences should be spell-checked or clarified for better understanding. Examples:
 "Whether naturally-occurring or engineered prokaryotic" comma before prokaryotic
 "The highest the CV the less homogeneous the population is." higher "Since just one NIa molecule per cell can cleave the target," meaning should be clarified.

The text has been revised and the suggested corrections done.

Reviewer #3

1) The authors mentioned that cells with digitalizer module showed obviously faster response after the induction of the system than non-digitalizer. The authors should make a further discussion about the mechanism behind this phenomenon.

In reality this feature is not captured by the model and varies depending on the expression system used (XylS/Pm or AlkS/PalkB). We have introduced a short discussion on this, we entertain that the dynamics of simple genetic devices can be affected by small changes in the interplay between the sRNA and its target RNA (Levine et al, 2007; Liu et al, 2011).

2) The authors used fixed inducer concentration in the validation experiments of the digitalizer module with GFP as reporter. A dose-response study of the digitalizer module would be more systematic.

The dose-response analysis has also been included for the digitalized circuit in Appendix Fig. S3.

3) In Figure 7D, it is interesting to notice that the on/off ratio of digitalizer module (pS239D•M) in *S. putida* is not as good as in *E. coli*. In *S. putida*, and there is a gap in on/off ratio between digitalizer and non-digitalizer. Is it because that the repression from LacI repressor is weaker in *S. putida*? Can the performance be improved by changing the repressor? It would be helpful if the authors could address this issue.

The levels of expression driven by the AlkS-PalkB system are much higher in *P. putida* than in *E. coli*, especially at the ON state. Therefore, the ON/OFF ratio is also higher for *P. putida*. However, the digitalizer module decreases the basal levels to very low amounts, as demonstrated for long time growth (24 h). Under these conditions, green fluorescence can be easily detected –even by the naked eye, unless the digitalizer is added to the expression system. Yet, indeed there is room for further improvement of the device in *P. putida*.

4) The presentation of some figures can be improved. Figure 1E and 4D are missing time(t) for the last column. Different fonts were used in one figure (Figure 3). Inconsistent graphic styles of subfigures (Figure 1D and 1E, Figure 4B and 4D) do not harmonize.

We have unified fonts, graphic styles and corrected figures 1E and 4D as suggested.

2nd Editorial Decision

27 September 2019

Thank you again for submitting your revised work to Molecular Systems Biology and apologies again for the delay in getting back to you. We have now heard back from the two referees who accepted to evaluate the revision. As you will see, the referees are globally supportive. Reviewer #2 raise however some remaining points, which we would ask you to carefully address in a revision of the present work. The main point refers to the need of a more careful use of the terms "bimodality" and "bistability" and the reviewer suggests the use of "switch-like", which could work. The reviewer also mentions that "digital" is strongly associated with "bimodality". I am not quite sure that this is

actually the case, and if clarified explicitly in the text, we would not be opposed to keep the term "digital".

REFEREE REPORTS

Reviewer #1:

I would like to thank the Authors for addressing my comments. I recommend the revised manuscript for publication in Molecular Systems Biology.

Reviewer #2:

The authors performed a hysteresis experiment with a negative result, indicating that bistability is unlikely to be present in the system. In order to indicate this change in the interpretation, the authors now use the term "bimodality". However, this does not seem to be a fortunate word choice for several reasons:

- a) The authors do not show experimental evidence that the distribution is bimodal; at least I could not find any distribution in the figures that indicates bimodality, a histogram with two peaks.
- b) In fact, it is better not to have bimodality since bimodality would counteract the main advantage of the presented system: the very low basal expression. Bimodality would imply that some cells in a population would have high expression, which the authors aimed to avoid.
- c) Bimodality has a complex relation with bistability and positive feedback loop. They are neither mutually exclusive nor exhaustive. Therefore, an appropriate term has to be found that correctly describes the system.

1) I suggest the term "switch-like" action of positive feedback. In terms of deterministic (ODE) modelling, a positive feedback can generate a switch-like or bistable response depending on whether ultrasensitivity is present or not. Stochasticity can induce bimodality in any of these scenarios. Since the authors detect neither bistability (hysteresis) nor bimodality, the switch-like action is the most likely feature of the feedback loop (PMID:27425609). It is unlikely that the ultrasensitivity can be directly measured with a fluorescence readout since the background fluorescence is relatively high in the system (Appendix Figure S3). The switch-like feature can be influenced (enhanced or suppressed) by cell growth (PMID:19801994). In this context, the authors can also discuss briefly that the total switch-like response is likely to be affected by growth (since the growth-related effect cannot not modelled adequately due its pleiotropic effects). The growth of induced and uninduced systems is measured in some but not all set-ups. I would suggest inserting a few additional sentences to indicate the growth rate differences in Figure S3 & S4.

2) Please avoid using the term "negative feedback" since the mutual repression is by definition positive feedback. The term "digital" is also strongly associated with bimodality. Thus, a different term would be more appropriate.

2nd Revision - authors' response

16 October 2019

Reviewer #2

The authors performed a hysteresis experiment with a negative result, indicating that bistability is unlikely to be present in the system. In order to indicate this change in the interpretation, the authors now use the term "bimodality". However, this does not seem to be a fortunate word choice for several reasons: (a) The authors do not show experimental evidence that the distribution is bimodal; at least I could not find any distribution in the figures that indicates bimodality, a histogram with two peaks. (b) In fact, it is better not to have bimodality since bimodality would counteract the main advantage of the presented system: the very low basal expression. Bimodality would imply that

some cells in a population would have high expression, which the authors aimed to avoid. (c) Bimodality has a complex relation with bistability and positive feedback loop. They are neither mutually exclusive nor exhaustive. Therefore, an appropriate term has to be found that correctly describes the system.

1) I suggest the term "switch-like" action of positive feedback. In terms of deterministic (ODE) modelling, a positive feedback can generate a switch-like or bistable response depending on whether ultrasensitivity is present or not. Stochasticity can induce bimodality in any of these scenarios. Since the authors detect neither bistability (hysteresis) nor bimodality, the switch-like action is the most likely feature of the feedback loop (PMID:27425609). It is unlikely that the ultrasensitivity can be directly measured with a fluorescence readout since the background fluorescence is relatively high in the system (Appendix Figure S3).

We agree that the system is not bistable, of course (this was handled in the previous revision). In this newly revised version we give in (not without some reluctance) to remove the term bimodality. However, the device has two different possible states (or modes): ON and OFF. OK, we may not call this bimodality, but we insist that this behaviour is *digital*. And a device with two possible states can be properly called a digital device—and the specific construct to bring this about, a *digitalizer*. In sum, in the revised ms. we adopt the suggested term switch-like. But we keep the term *digital*, as it captures the essence of the device.

The switch-like feature can be influenced (enhanced or suppressed) by cell growth (PMID:19801994). In this context, the authors can also discuss briefly that the total switch-like response is likely to be affected by growth (since the growth-related effect cannot not modelled adequately due its pleiotropic effects). The growth of induced and uninduced systems is measured in some but not all set-ups. I would suggest inserting a few additional sentences to indicate the growth rate differences in Figure S3 & S4.

Thanks for the comment. We have added a few sentences in the text to highlight these possibilities

2) Please avoid using the term "negative feedback" since the mutual repression is by definition positive feedback. The term "digital" is also strongly associated with bimodality. Thus, a different term would be more appropriate.

The earlier texts re *negative feedback* have been edited to improve clarity. However, we would still like to keep the term *digital*, as it evokes the states zero-to-full that are displayed by the device presented in the work.

Accepted

24th October 2019

Thank you again for sending us your revised manuscript. We are now satisfied with the modifications made and I am pleased to inform you that your paper has been accepted for publication.

Corresponding Author Name: DE LORENZO

Journal Submitted to: MSB

Manuscript Number: MSB-18-8777R